# 3D Bioprinting of Hyaline Articular Cartilage: Biopolymers, Hydrogels, and Bioinks

**DOI:** 10.3390/polym15122695

**Published:** 2023-06-15

**Authors:** Larisa T. Volova, Gennadiy P. Kotelnikov, Igor Shishkovsky, Dmitriy B. Volov, Natalya Ossina, Nikolay A. Ryabov, Aleksey V. Komyagin, Yeon Ho Kim, Denis G. Alekseev

**Affiliations:** 1Research and Development Institute of Biotechnologies, Samara State Medical University, Chapayevskaya St. 89, 443099 Samara, Russiavolovdm@mail.ru (D.B.V.);; 2Skolkovo Institute of Science and Technology, Moscow 121205, Russia; i.shishkovsky@skoltech.ru; 3RokitHealth Care Ltd., 9, Digital-ro 10-gil, Geumcheon-gu, Seoul 08514, Republic of Korea; yeonho.kim@rokit.co.kr

**Keywords:** hyaline articular cartilage, traumatology and orthopaedics, regenerative medicine, tissue engineering, biofabrication, three-dimensional bioprinting, bioprinters, scaffolds, biopolymers, hydrogels, bioinks, cells, signal molecules

## Abstract

The musculoskeletal system, consisting of bones and cartilage of various types, muscles, ligaments, and tendons, is the basis of the human body. However, many pathological conditions caused by aging, lifestyle, disease, or trauma can damage its elements and lead to severe disfunction and significant worsening in the quality of life. Due to its structure and function, articular (hyaline) cartilage is the most susceptible to damage. Articular cartilage is a non-vascular tissue with constrained self-regeneration capabilities. Additionally, treatment methods, which have proven efficacy in stopping its degradation and promoting regeneration, still do not exist. Conservative treatment and physical therapy only relieve the symptoms associated with cartilage destruction, and traditional surgical interventions to repair defects or endoprosthetics are not without serious drawbacks. Thus, articular cartilage damage remains an urgent and actual problem requiring the development of new treatment approaches. The emergence of biofabrication technologies, including three-dimensional (3D) bioprinting, at the end of the 20th century, allowed reconstructive interventions to get a second wind. Three-dimensional bioprinting creates volume constraints that mimic the structure and function of natural tissue due to the combinations of biomaterials, living cells, and signal molecules to create. In our case—hyaline cartilage. Several approaches to articular cartilage biofabrication have been developed to date, including the promising technology of 3D bioprinting. This review represents the main achievements of such research direction and describes the technological processes and the necessary biomaterials, cell cultures, and signal molecules. Special attention is given to the basic materials for 3D bioprinting—hydrogels and bioinks, as well as the biopolymers underlying the indicated products.

## 1. Hyaline Articular Cartilage Damages, Regeneration Process, and Current Treatment Approaches

Hyaline articular cartilage (HAC) is a smooth, wear-resistant, highly specialized hyaline cartilage covering the epiphyses and certain anatomical areas of the bone within the synovial joint capsule (Figure 1). HAC reduces friction, allowing smooth joint movement [1,2]. The HAC lacks nerve endings as well as blood vessels, so its nutrition depends on articular (synovial) fluid, the underlying (subchondral) bone, and mechanical loading [3,4]. Through the porous upper layer of the cartilage matrix, nutrients soluble in synovial fluid enter, and metabolism products are removed from HAC. With the underlying subchondral bone, metabolism is realised by diffusion from numerous blood capillaries [5,6].

HAC is laid in the foetal period in which its regeneration is possible. After birth, there is moderate regenerative activity in the HAC in childhood. In adults, the HAC is not regenerated, except if injuries affect the subchondral bone, when regeneration is possible due to the proliferation of poorly differentiated osteogenic cells located in the zone of the blood capillaries of bone tissue, and their subsequent differentiation in two directions: into bone and cartilage cells. In such cases, the regeneration process is not comprehensive because fibrous cartilage tissue is formed, the mechanical properties of which are worse than the hyaline one [7,8,9].

Macroscopically, HAC has a homogeneous, opalescent appearance with a bluish tint. Morphologically, HAC is a special connective tissue consisting of cells (chondroblasts and chondrocytes) and an extracellular matrix (ECM), which they produce. The ECM is formed by glycosaminoglycans, glycoproteins, collagen and elastin fibres, and water [3,10,11].

In HAC, chondrocytes are located in the lacunas and completely fill them. Chondrocytes synthesize and secrete all components of the surrounding ECM. In the adult organism, chondrocytes do not divide because they are long-lived and age with the body. Chondroblasts, or perichondrial cells, are mesenchymal progenitor cells that form chondrocytes in the growing ECM as a result of endochondral ossification. Chondroblasts are the youngest cells in the HAC and they are capable of mitosis [12,13,14].

The predominant biopolymer of ECM is collagen type II. Collagens of types IX, X, and XI are identified in small amounts. The tensile strength of cartilage is conditioned by collagen. As we age, the water content of cartilage and the number of bonds between collagen molecules gradually decrease. As a result, cartilage tissue becomes less elastic and has less resistance to stretching, twisting, and compression loads. In other words, as we age, the cartilage becomes more vulnerable to damage [15,16,17,18].

Glycosaminoglycans are also identified in significant amounts in the cartilage ECM. They form macromolecular aggregates that bind water. Cartilage owes its resistance to pressure to the ability of ECM glycosaminoglycans to attract and retain water. The most characteristic glycosaminoglycans for HAC are chondroitin sulfate and keratan sulfate. The core of the structure of glycosaminoglycans is a giant molecule of such biopolymer as hyaluronic acid. Large molecules of the glycoprotein chondronektin also control the consistency of the surrounding ECM [19,20,21,22].

The ECM of living cartilage contains macromolecule-bound water, which provides elasticity to cartilage tissue and diffusion. The latter is the only way for nutrient and hormone intake into the chondrocytes, as well as metabolite removing and gas exchanging [10,23].

In general, the HAC is a highly organized striated structure in which cells (chondrocytes and chondroblasts) and ECM components, such as collagen fibres and glycosaminoglycan molecules, are arranged in a strict order, depending on the depth of the cartilage zone. Anatomically, four such zones are distinguished:Superficial;Transitional;Deep;Calcified.

Each zone has a different composition of the ECM, structural organization, and cell density. Such a unique anatomical structure of the HAC determines the gradient of its physical, mechanical, and biological properties. Additionally, the synergistic functionality of the HAC and subchondral bone is very important. The bone–cartilage interface in joints is formed by different layers of bone and cartilage cells with a gradient of mechanical properties and cell organization [7,9,10,24].

The complex striated structure of the HAC, and the peculiarities of its trophic (absence of supracartilage as well as feeding vessels and nerves), cause constrained self-regeneration of the HAC [14,25,26,27]. Therefore, its destruction due to traumatic injuries or pathological processes, such as osteoarthritis and rheumatoid arthritis, is an actual problem, complemented by the wide prevalence of the pathology in the population [28,29,30,31,32].

Minor defects of the HAC after the slight injuries or in the early stages of a disease can be treated with physical therapy and medication. In the latter variant, along with anti-inflammatory and metabolic systemic therapy, local treatment is also important. An example is viscosupplementation or injection of hyaluronic acid preparations into the joint cavity for normalizing the intra-articular environment, as well as to restore such properties of synovial fluid as elasticity and viscosity. It has analgesic, anti-inflammatory, anabolic, and chondroprotective effects and stalls the progression of the disease. The effectiveness of viscosupplementation depends largely on the type of medication, its origin, production technology, and physical and chemical properties [33]. Cross-linked hyaluronic acid hydrogels have the most acceptable characteristics today [34,35,36]. However, as the defects progress, significant cartilage destruction occurs, which worsens the patient’s quality of life due to severe pain, stiffness, and swelling in the affected joints [31,37,38,39,40].

Current methods of conservative treatment of HAC injuries include anti-inflammatory, analgesic, and osteochondrometabolic pharmacotherapy, as well as weight-bearing and physical therapy. Such treatment settings allow, up to a point, relief of the clinical symptoms listed above, but they cannot repair the HAC or stop its destruction [41,42,43]. Therefore, in more severe and neglected cases, surgical intervention is required [44,45]. Chondroplasty is the most widespread approach in reconstructive surgery for HAC. Actually, it is performed predominantly arthroscopically, under video control, using minimally traumatic endoscopic instrumentation [46,47]. Chondroplasty options with proven efficacy include such interventions as:Subchondral tunnelling;Microfracturing;Osteochondral autotransplantation;Allo-osteochondroplasty.

The above interventions are widely used in clinical practice but also have drawbacks, such as the trauma of the donor site, hyper- and hypotrophy of the graft, death of chondrocytes, and resulting incomplete reparation of HAC with the formation of fibrous cartilage areas, preservation of local defects, and increase in their size and depth [48,49,50].

As a result, with progress of the disease, and irrational approaches to treatment, significant destruction of the HAC happens, which requires a total joint replacement [51,52]. However, this operation has severe limitations and contraindications, and therefore may not be performed in all needy patients [53,54]. In addition, endoprosthetics is a “one-way ticket” without the possibility of restoring the native elements of the joint. Further, in the event of various individual reactions to the parts of the endoprosthesis; the development of complications, such as instability of the artificial joint; or periprosthetic infection, the patient is “doomed” to a technically more difficult and traumatic operation of re-endoprosthetics or a disabling intervention of arthrodesis of the joint [55,56,57,58,59].

Thus, at the present stage, there is a need to develop new approaches to the treatment of HAC pathologies, capable of effectively providing regenerative replacement of cartilage defects and their subsequent functionalization.

## 2. Regenerative Medicine and Tissue Engineering in the Treatment of Hyaline Articular Cartilage Injuries

New perspectives in solving the problem of HAC injuries for scientists and doctors turned out to be related to biomedicine and its advanced direction such as regenerative medicine and corresponding to its molecular–biological, cellular, tissue-engineering, and other closely related and interrelated fields of scientific research. At the heart of this field is the process of regeneration, which is the ability of living organisms to renew (physiological regeneration) or, over time, to repair damaged tissues and sometimes entire lost organs (reparative regeneration) [60,61,62,63].

Learning how to manage these capabilities means improving the quality of human life, prolonging working age, and reducing the cost of long-term treatment of patients with chronic diseases. The main exploratory directions of regenerative medicine are:Stimulation of regeneration with bioactive factors accelerating cell reproduction, growth, and differentiation;Cell therapy using stem cells;Tissue engineering.

Thus, the main areas of research in regenerative medicine are related to the use of factors affecting cell growth and maturation; cell and tissue engineering, which is a new biomedical discipline involving the use of a combination of cells; and biomaterials and suitable biochemical and physicochemical factors, as well as engineering approaches, to restore, maintain, improve, or replace various types of biological tissue [64,65,66,67].

## 3. 3D Tissue Bioprinting

One of the promising areas of tissue engineering is biofabrication, which specializes in the research, development, and implementation of biologically modified processes and automation in the production of functional tissue and organ analogues. At the same time, the creation of bioanalogues occurs in vitro, through bioassembly, three-dimensional (3D) bioprinting (TDB), and several other methods, such as directed assembly, enzymatic assembly, and self-assembly and the subsequent processes of functionalization (“maturation”) of tissues [68,69,70,71,72,73,74]. A wide range of sources, including biomaterials of various origins and their derivatives, signalling molecules, and cells and their aggregates, are used to create biobased products [75,76,77,78].

Many approaches of automated assembly of tissue-engineered constructs by TDB methods were a qualitatively new step in biofabrication and a separate direction in 3D printing technology, whose emergence in the late 20th century paved the way for mainstream innovations in many areas, such as engineering, industry, art, education, and medicine [73,74,79,80,81,82]. In modern 3D printers, cartridges with print heads can move in three dimensions during the volume printing process and distribute various materials—polymers, metals, ceramics, and even chocolate—in space, forming, layer-by-layer, 3D objects [83,84]. The combination of technologies that create a 3D object by adding material in a layer-by-layer manner is called additive manufacturing [85,86,87,88].

In medicine, additive manufacturing technologies are used to manufacture disposable sterile instruments, including personalized instruments for a specific patient; to create individual dental structures, prostheses, and crowns in dentistry; for hearing aids and implants in otorhinolaryngology; in traumatology and orthopaedics—for printing prostheses, endoprostheses, and orthoses, taking into account anatomical features of patients; in 3D printing of dummies (phantoms) and organ models for the educational process; and for microfabrication, which allows for printing medical devices and parts of micron-sized instruments [80,85,89,90,91]. A qualitative breakthrough in 3D printing technology, which occurred at the beginning of the 21st century, allowed scientists to print volume constructs using biomaterials, living cells, and auxiliary components, and to further create on their basis fully functional analogues of living tissues and organs [92,93].

Compared to conventional 3D printing, there are some factors that complicate the production process in TDB. These include the proper selection and combination of biomaterials, cells, and signal molecules, as well as consideration of the technical complexities associated with the equipment used. Such issues require the interaction of technologies at the intersection of engineering, physics, and biomedicine. From a technical point of view, the TDB process involves three sequential steps: pre-bioprinting, bioprinting, and post-bioprinting [82,93,94].

Pre-bioprinting, or the “preparation” stage, includes computer modelling of the future 3D object; isolation and cultivation of cell cultures; and biomaterial adjustment.Bioprinting, or the “production” stage, involves the creation of a volume tissue-engineered construct in a 3D bioprinter by “layer-by-layer” deposition of biomaterials, auxiliary components, and living cells on a substrate.Post-bioprinting, or the “functionalization” stage, is necessary for the stabilization of the bioprinted construct and “maturation” of its cells; this stage is implemented in bioreactors, where basic structural and functional characteristics of a bioprinted construct such as mechanical strength, structural integrity, and others are formed.

Thus, preparing and implementing the TDB process and capturing printed products requires, at a minimum, the following critical components [93,95,96]:

Equipment: 3D scanner (appropriate medical diagnostic equipment—MSCT, MRI, and 3D X-ray), personal computer (specialist workplace), 3D bioprinter, cell and tissue bioreactors, and equipment for input and output certification.3D printing job preparation software, CAD software for product design, and special utilities for converting from the DICOM format (MSCT and MRI data storage format) to 3D printing data format (STL files).Cell culture, as well as biomaterials of a natural or synthetic origin (including their combinations), as the basis of the volume matrix (analogue of ECM), in which cells will be placed.

### 3.1. The Pre-Bioprinting Stage

#### 3.1.1. Visualization and 3D Modelling

The process of creating 3D tissue-engineered constructs begins with visualization and 3D modelling at the pre-bioprinting stage and proceeds as follows: when designing a future printable object, the bioengineer sets the dimensions, geometry, number of layers, and other characteristics of the model according to the tasks to be performed or based on data obtained by magnetic resonance imaging (MRI) or multi-slice computed tomography (MSCT) and consultation with medical personnel [96,97,98]. Accordingly, the design of the TDB object can take place as follows:Based on the data obtained by MRI or MSCT of the intervention area (recipient area).De novo, when a bioengineer uses 3D bioprinter software to create a model and set the conditions for the location of the final product in the body.Meanwhile, the following tasks are solved [93,97] (Figure 2):Computer processing (transformation) of data from different sources (MSCT, MRI, 3D X-ray, CAD systems, or CAD-design) by special programs to create a digital model of the future 3D product; its location in the synthesis chamber, splitting into layers; and formation of support (if necessary).Consultation with medical personnel (surgeon and traumatologist) on the correctness of the digital model and its compatibility with the course, scope, and sequence of the planned operation. Biomaterial adjustment with the method and mode of the 3D bioprinting definition according to the tasks to be performed.3D printing of a pre-product with usage of convenient and cheap material for visual, tactile, and volume–geometric diagnostics, as well as pre- and post-operative planning.

Further, the created digital 3D models should be saved and stored in secure and closed databases (blockchain technologies). In appropriate cases, digital 3D models are retrieved from databases using 3D CAD to analyse or control the results of surgical intervention, and to apply in other similar cases [99,100].

#### 3.1.2. Preparation of Cell Culture

The main tasks at the pre-bioprinting stage are the adjustment and preparation of cell cultures, biomaterials, and auxiliary components for the biofabrication of tissue-engineered constructs at the subsequent stage of bioprinting.

Cells are one of the main components of hybrid cell–tissue constructs and, consequently, the TDB process. Preparation of cells for bioprinting involves the preparation of the donor, then fence of cells containing biological material, isolation, and cultivation of primary cell cultures with subsequent formation of cell spheroids from cell suspensions. A spheroid is a tiny (200–300 microns) ball-shaped clot of living cells (Figure 3). The use of cells for TDB in the form of spheroids is due to the high density of cells, their synthesis of ECM, and the ability to merge into organ-like structures with the formation of tissue architectonics. Cells in spheroids have higher resistance to stress, radiation, and other adverse factors. A big plus is the possibility of automated production of spheroids of different diameters containing different types of cells. In particular, researchers have created cell spheroids for biofabrication and TDB of HAC biosimilars, from mesenchymal stromal cells (MSCs), fibroblasts, chondroblasts, osteoblasts, and chondrocytes, as well as their combinations [24,70,101,102,103,104].

#### 3.1.3. Adjustment of Biomaterial and Printing Technology

Biofabrication of tissue analogues using TDB technology requires not only cells but also biomaterials. The latter is (as defined by IUPAC, 2012) synthetic or natural materials used in contact with living biological tissues, including as part of medical devices [105]. One of the main requirements for all biomaterials is biocompatibility. It refers to the ability of performing a specific function for the required time without harming the surrounding tissues and the whole body [103,106,107]. In some cases, biomaterial products should easily undergo biodegradation (resorption). Biodegradation is characterized by a decrease in the size and weight of a biomaterial product (biodegradation process) during its functioning under the influence of components of the surrounding biological environment. The areas of biodegradation are replaced by the recipient’s surrounding tissue [108,109,110]. In addition, biomaterials can be biomimetic, which means reproducing of structure and functions of a tissue or organ [111,112].

Biomaterials are used to form an analogue of the ECM, which can contain auxiliary components and cells of the future construct and, in the beginning, be a source of nutrition for the cells, contributing to their growth and proliferation. The type of biomaterial dominantly determines the technique and modes of TDB [113,114]. Next, basic approaches to the formation of volume tissue-engineered constructs can be distinguished according to the nature of the ECM’s analogue formed from biomaterials, in which cells and auxiliary components will be placed [115,116]:With the formation of scaffolds;Without scaffold formation or scaffold-free;Hybrid.

In regenerative medicine, a wide range of biomaterials, ranging from mechanically strong but relatively inert biosilicates to bioactive hydrogels, and their combinations are used to repair defects of the HAC [102,117,118,119].

#### 3.1.4. Bioprinting with the Formation of Scaffolds

Historically, TDB in the context of traumatology and orthopaedics began with scaffold formation techniques. Such techniques of TDB vary depending on the materials used and the corresponding scaffolding methods. Traditionally, the following biomaterials are applied to form scaffolds [120,121,122,123,124]: Metals;Silica, including biosilicates;Hydroxyappatite and other calcium phosphates;Alumina;Polymer clays, including laponite;Nonresorbable synthetic biopolymers;Combinations of the aforementioned materials.

Additionally, the following techniques are used to create 3D objects [125,126,127,128,129].

Powder methods of 3D printing with biocompatible metals and alloys:Laser (E-beam) Powder Bed Fusion (L/E-PBF);Direct Energy Deposition (DED).3D printing with liquid materials (photobiopolymers):Vat Photo Polymerization: Stereolithography (SLA, including two-photon) or Digital Light Processing (DLP);Ink Jet Printing (IJP).3D printing with solid materials:Fused Deposition Modelling (FDM);Binder Jetting (BJ).

The ideal scaffold must reproduce the unique mechanical and biological properties of the native ECM of the desired tissue and have a porous structure that allows cells to attach, signalling molecules and nutrients to spread and vessels and nerve fibres to sprout [130,131,132]. Magnesium alloys and simple organic compounds can be applied to make mechanically solid and long-lasting structures. Both magnesium and calcium compounds have pronounced chondrogenic and osteogenic effects, and phosphate and silica ions are important biologically active agents capable of enhancing bone and cartilage differentiation [132,133,134,135]. Bioglass-based scaffolds (70S30C, S53P4) can also stimulate the maturation and differentiation of supporting tissue cells, due to a composition that includes silica ions and calcium phosphate, other auxiliary components [136,137,138,139]. Moreover, the chondroosteogenic potential of bioglass scaffolds can be potentiated by incorporating metal ions such as strontium, magnesium, and zinc [140,141,142].

However, scaffolds have a serious disadvantage, namely brittleness, since the structure is mechanically rigid but unplastic and prone to significant fracture under compression [143,144]. A solution may be the use of auxetic materials [145]. Another disadvantage is that the process of creating scaffolds involves exposure to high temperatures or aggressive solvents (hardeners). Some synthetic polymers (methyl methacrylate and polyurethane) have a toxic polymerization process. Therefore, scaffold TDB is technologically incompatible with synchronous inoculation of live cell cultures. The latter are inoculated into the scaffold or seeded on its surface after the fabrication of the construct to avoid exposing the cells to adverse factors [146,147]. Thus, the biofabrication of hybrid cell–tissue constructs using scaffolds occurs in several successive stages, which can be attributed to the disadvantages of this method of TDB.

#### 3.1.5. Scaffold-Free Bioprinting

In HAC biofabrication, the TDB method without the formation of scaffolds, but using the following materials to form volume constructs, has been more widely used [72,148,149,150]:Biopolymer hydrogels without living cells or so-called biopaper;Cell-loaded (with spheroids) biopolymer hydrogels or so-called bioinks.

##### Biopolymer Hydrogels

Hydrogels are a variant of colloidal gels—they are a network of tridimensional cross-linked hydrophilic biopolymer chains, which, due to their structure (presence of hydrophilic groups), can swell and retain water in a volume exceeding their original mass (up to 90%) and are penetrated deep into tissues; in doing so, they maintain their structural integrity [151,152]. This circumstance is due to the structural similarity of hydrogels with natural HAC. Thus, it is the hydrogels that have the good potential in the context of regenerative medicine for lesions of HAC [153,154].

Hydrogels are applied to create the basis of a future hybrid cell–tissue construct, i.e., an analogue of ECM. It allows organizing the conditions necessary for cell growth, tissue formation, and providing various combinations of components within the hydrogels with the desired combinations of properties of the final product [155,156,157]. In addition, hydrogels help scientists to create, study, and model the various interactions of the cells introduced into the hydrogel, as well as to evaluate the effect of hydrogels, as a cellular matrix, on the growth, differentiation, and proliferation of the latter [151,157,158,159].

Hydrogels have unique distinctive properties, such as biocompatibility, non-toxicity, biodegradability, and a lack of carcinogenicity, immunogenicity, and other irritating effects on tissue [160,161,162,163]. Additionally, we can note the ability to swell, high flexibility, elasticity, and the possibility to modify the given mechanical, physical, chemical, biological, and morphological qualities [161], which, taken together, allow the active application of hydrogels in the practical application of TDB [154,164,165,166,167,168]. Physical and chemical properties of hydrogels also depend on their biopolymer composition, cross-linking methods, and the strength of the bonding between biopolymer molecules [34,155,156,157] (Table 1). The mass percentage and molecular weight of the biopolymer determine the viscosity of the hydrogel, which also affects the mechanical properties and degree of cross-linking of the latter and the accuracy of the TDB process [164,165,166,167,168,169,170].

The structure of hydrogels is maintained by chemicals, including temporal bonds (ionic, covalent, weak, and strong) between biopolymer molecules, which are hydrophilic. The hydrophilicity provides hydrogels with the ability to interact with water and absorb it. This leads to the plasticization of long chains of biopolymers, which makes them suitable for applying at physiological human and animal body temperatures. The latter circumstance is actual for the cell proliferation [166,168,169,172,173].

During gelation, macromolecular chains are joined together, initially resulting in a gradual increase in branched but soluble biopolymers, depending on the structure and conformation of the starting material [171]. Gradually, the binding process increases the size of the branched biopolymer with a decrease in solubility, which ultimately contributes to the formation of a hydrogel [171].

According to the reversibility of the gelation process, all hydrogels may be categorized into two groups [171]:Irreversible or chemically cross-linked hydrogels are covalently “cross-linked” networks of biopolymers in which strong and more stable covalent bonds replace hydrogen bonds [161,174]. In such hydrogels, the equilibrium swelling state depends on the biopolymer–water interaction parameter and the density of covalent bonds [161,174,175].Reversible or physical hydrogels—in these, complex networks of biopolymers are held together by ionic, hydrogen, and hydrophobic interactions and other physical interactions that exist between different polymer chains [161,174].

Methods of hydrogels’ obtaining are premised on the principles of physical or chemical cross-linking, i.e., binding of the biopolymers’ molecules together [168,171,176]. The method of physical cross-linking [161,177] is simple and makes it possible to form a wide range of hydrogel textures by selecting the optimal biopolymer, its concentration, and its pH. Varieties of this method include:A method of ionic interaction in which ionic biopolymers are cross-linked by adding divalent or trivalent counterions, for example, sodium alginate with calcium chloride divalent [161].The method of complex coacervation. Complex coacervation hydrogels can be formed by mixing a polyanion with a polycation, for example, the coacervation of polyanionic xanthan with polycationic chitosan [161,178].Method of hydrogel formation with hydrogen bonds. A hydrogel is obtained by lowering the pH of an aqueous solution of biopolymers carrying carboxyl groups. For example, sodium in carboxymethyl cellulose (CMC) is replaced with hydrogen in acid. A hydrogen-bonded network is formed by dispersing CMC in a 0.1 M hydrochloric acid solution [161,179,180].Maturation method (aggregation by heat, light, pressure, and mechanical action). The aggregation of protein components caused by heat treatment increases the molecular weight and subsequently leads to a hydrogel form with improved mechanical properties and the ability to bind water [161,171].Freeze–thaw method. As an example, the formation of microcrystals in the biopolymer (polyvinyl alcohol and xanthan gels) structure due to a freeze–thaw [161,176,181,182]. Hydrogels formed by this method have sufficient mechanical strength and stability but are opaque in appearance and have a low ability to swell [161,176,181].The method of heating–cooling a biopolymer solution involves the use of a temperature factor. Cross-linking of the hydrogel occurs when hot solutions of gelatine or carrageenan are cooled due to the formation of a helix of molecules, combining the helixes and forming bonding zones [161,182].Isostatic ultrahigh pressure method (IUHP-method), in which solutions of natural biopolymers, such as starch, are exposed to an ultrahigh pressure of 300–700 MPa for 5 or 20 min in a chamber, resulting in changes in polymer morphology [161,177].Radiation cross-linking method. This method is quite common, does not require the addition of chemical agents, allows for combining modification with sterilization, and, at the same time, preserves the biocompatibility of the polymer. The method is premised on the releasing of free radicals in the polymer after exposure to an electromagnetic radiation source (gamma rays and X-rays) [161,183].

The method of chemical cross-linking is realised by incorporating monomers into the main chain of biopolymers or by using a cross-linking agent to bind the two biopolymer chains. Cross-linking of natural and synthetic biopolymers can be achieved by reacting their functional groups (such as -OH, -COOH, and -NH) with cross-linking agents such as aldehyde (e.g., glutaric aldehyde and adipic acid dihydrazide) [161,177,184]. Variations of this method include:The method of chemical cross-linking agents’ usage. Involves introducing new molecules between polymer chains to produce cross-linked chains, such as glutaric aldehyde, and epichlorohydrin [161].The method of monomer polymerization is based on a preformed biopolymer. Polymer chains are activated by the interaction of chemical reagents, whereby the growth of functional monomers leads to branching and further cross-linking of the hydrogel [161].The method of hydrogels’ forming using gelling agents such as glycophosphate, 1-2-propanediol, glycerol, trehalose, mannitol, etc. [185,186].

There are a huge number of biopolymers of a natural and synthetic origin from which hydrogels are formed using the methods listed above [155,157] for use in regenerative medicine. Biopolymers of a natural origin include hyaluronic acid [150,187,188], chitosan [150,187,189], alginate [150,168,190], agarose [150,191], gelatine [150,192], collagen [150,193,194], heparin [195,196], fibrin [197,198], etc. The group of synthetic polymers includes polyvinyl alcohol, polyethylene glycol, sodium polyacrylate, acrylate polymers, and their copolymers [150,199], as well as polyurethanes, poly(propylene fumarates), polyphosphazenes, polyethylene glycols, and others [152].

Despite the wide range of available biopolymer compounds, hydrogels made of biopolymers of a natural origin have potential advantages over those made of synthetic sources. These include better biocompatibility, more physiological resorption, and low immune response [157]. The properties of hydrogels can be adjusted by combining synthetic biopolymers with natural ones, for example, by their chemical modification [200]. Currently, there is an active search for new polymeric materials that can be used to manufacture advanced hydrogels with given properties [200,201].

Natural biopolymers used to create hydrogels, in turn, can be divided into allogeneic, xenogeneic, and autologous according to their origin and relation to the recipient tissues. Several biopolymers, such as hyaluronic acid, gelatine, and collagen, can simultaneously belong to all the groups above since their sources can be both animal material (xenogenic) and human material (autologous and allogenic, obtained intraoperatively or posthumously). Characteristics of some basic biopolymers of a natural origin used as a base for hydrogel fabrication are presented below.

Hyaluronic acid is a structurally simple polymeric glycosaminoglycan consisting of repeating links of β-1,4-D-glucuronic acid-β-1,3-N-acetyl-D-glucosamine residues [150,152]. It is a major component of the HAC’s ECM, where it plays a crucial role in maintaining homeostasis by regulating cellular functions, including the promotion of the chondrogenic phenotype, as well as the creation and maintenance of ECM components [187]. The advantages of using this biopolymer are its excellent biocompatibility and bioactivity, as well as a significant stimulation of angiogenesis in humans [202]. Hyaluronic acid-based hydrogels improve cellular functionality by enhancing their synthesis of ECM elements, increasing the expression of chondrogenic gene markers, and, consequently, accelerating cartilage formation [32,34]. The disadvantages of hyaluronic acid-based hydrogels are their insufficient strength and slow solidification, which limits their application in the aspect of TDB [188].

Sodium hyaluronate is a mucopolysaccharide, one of the components of mammalian connective tissue. Sodium hyaluronate consists of macromolecules, which, in turn, consist of disaccharide links linked by N-acetyl-D-glucosamine and D-glucuronic acid molecules. Hydrogels based on it are characterized by the same advantages and disadvantages as those based on hyaluronic acid.

Sodium alginate is an ionogenic polysaccharide obtained by the alkaline extraction of brown algae [150,152]. Alginates are salt derivatives of alginic acid and are represented by long chains of polysaccharides, which give structure plasticity and gel-forming ability. This biopolymer is widely used in medical practice in the aspect of hydrogel synthesis, due to its zero toxicity, good water solubility, and swelling ability coupled with its tendency to thicken [151,203,204]. Alginate in aqueous solutions forms hydrogels when bivalent metal salts (calcium) are added as cross-linking agents, which interact with the carboxyl groups of hyaluronic units of alginate molecules [151,203,204]. Alginate hydrogels are frequently used as a base of bioinks for TDB [190]. Such bioinks can modulate the proliferation and migration of allogeneic and autologous adipose mesenchymal stem cells (hADSCs) without disrupting their structural integrity [172,190,205].

Collagen is a natural protein polymer most commonly found in the ECM of both dense and loose connective tissues [150,152,206,207]. Collagen maintains the structural and biological integrity of the ECM [152,188], morphogenesis, and cellular metabolism of tissues, giving them the required mechanical and biochemical properties [152,205,207]. A collagen molecule is a right-handed spiral of three α-chains with three amino acid residues in each turn of the spiral. The structural unit of collagen is a tropocollagen. Rows of tropocollagen molecules are the basis of the structural organization of collagen fibrils. The formation of collagen fibres occurs through the aggregation of microfibrils. Fibril formation occurs spontaneously by self-assembly [208]. At the same water content, the denaturation temperature of fibrillar collagen is 65 °C and that of tropocollagen is 40 °C [209]. Due to the content of arginine–glycine–aspartate sequences in the primary structure, collagen is a biomaterial that provides cell adhesion [209,210,211]. To date, about 28 types of collagen have been identified in which a triple helical tertiary structure is present [205,207], among which collagen type I is the most common type found in ECM, especially in tissues such as tendons and bone [205,207,212]. Collagen plays a significant role in morphogenesis and cellular metabolism of new tissues, giving them the required mechanical and biochemical properties [188,205,207].

Collagen has considerable potential for use as a biomaterial for tissue engineering because of its availability, biocompatibility, ease of combination with other materials, ease of processing, hydrophilicity, low antigenicity, high porosity of formed structures, good biodegradability in the body, etc. [200,201,207,213]. Collagen has haemostatic properties, so it is applied in the manufacturing of artificial valves and vessels [214,215]. When injected into the body, collagen stimulates reparative processes, contributing to the formation of its own collagen, but does not provide complete regeneration of the organ, resorbing earlier and forming scar tissue [216,217]. Unregulated and rapid biodegradation time (up to 1 month) significantly reduces the functioning period of collagen products. The formation of a heterogeneous supramolecular structure of collagen-containing gel makes it possible to slow down biodegradation [216,218]. Collagen may also be combined with other biopolymers, such as alginate [219].

Different types of collagens are used in bioengineering, both of a xenogenic and allogenic origin [193,194]. Collagen types I, II, III, V, and IV are most often applied in tissue engineering. Among them, collagen type I is considered to be one of the most valuable, because it contains up to 90% of the connective tissue protein [207,217,220]. In the hydrogel state, collagen exists under certain conditions, such as a neutral pH environment and physiological temperature [220,221]. Collagen type II is the most common collagen in HAC and constitutes about 90–95% of its ECM, and the remaining volume is accounted for by collagens of types VI, IX, X, and XI [166,222]. Collagen types II, IX, and XI form a network of fibrils, which traps macromolecules and maintains the tensile strength of the ECM [166,222]. Collagen type II has some advantages over type I collagen due to its availability and cost, better compatibility due to less antigenic activity, and higher mechanical strength of the structures formed [166,223]. In addition, in a hydrogel based on type II collagen, MSCs differentiate more effectively into chondrocytes compared to a hydrogel made of type I collagen [166,224].

Gelatine is a protein product whose main component is denatured or hydrolysed collagen, which is the main component of the ECM in most tissues [225,226]. Under natural conditions, collagen is in the form of long fibrillar filaments, but when heated, these filaments turn into disorderly balls [150,152,161,227]. Gelatine contains a lot of arginine–glycine–aspartic acid sequences that can fix cells [152,157,158], as well as target sequences of matrix metalloproteinases suitable for cell remodelling [228,229,230,231,232,233,234]. Compared with collagen, the advantages of gelatine are better solubility and less antigenicity [157,158,228,229,230]. In addition, the gelatine solution has the unique attribute, namely gelation at physiological temperatures, forming physically cross-linked hydrogels [150,152,157,231,232,233,234]. Gelatine is used to prepare hydrogels and bioinks with different initial concentrations of the biopolymer [192]. However, all formulations with gelatine exhibit the required rheological and mechanical qualities as well as swelling abilities, suitable for TDB and cell encapsulation [192]. In addition to its single use, gelatine is also applied to product hydrogel mixtures with agarose [192]. A recent study showed that SH-SY5Y cells differentiated into neuron-like cells using developed gelatine–agarose hydrogel-based bioinks, exhibiting high viability (>90%) after 23 days in culture [192]. This study demonstrates the preferential properties of the gelatine–agarose mixture in terms of creating hydrogels and bioinks for TDB [157,192,229,231,233].

Chitosan is a partially or fully deacetylated derivative of the natural polysaccharide chitin (poly-1-4-N-acetyl-2-amino-2-deoxy-β-D-glucopyranose) [152,235]. Chitosan is applied in the production of hydrogels and bioinks for TDB and is superior to alginate in its ability to stimulate cell proliferation and differentiation. Previously, scientists have demonstrated that cells inside a chitosan-based hydrogel mineralized and osteogenically differentiated after 21 days of cultivation [187,189].

Agarose is a polysaccharide derived from some red algae and consists of β-1,3-linked-D-galactose and α-1,4-linked 3,6-anhydro-L-galactose links [152,188,236]. Agarose is one of the two main components of agar. Standard agarose derived from Gelidium algae has a gelation temperature of 34–38 °C, while agarose derived from Gracilaria algae has a gelation temperature of 40–52 °C [237]. The elastic modulus of agarose ranges from one to several thousand kPa depending on polymer concentration and molecular weight [238]. Agarose forms a reversible hydrogel, and the mechanism of agarose gelation is the formation and aggregation of double helixes due to intermolecular hydrogen bonds upon cooling [236,239]. It is also applied as a basic component of hydrogels and bioinks for TDB of supporting tissue analogues because of its homogeneity and adequate mechanical properties. Moreover, agarose is cytocompatible and allows the preservation of cell morphology [150,240,241].

Fibrinogen is a precursor of fibrin, which is isolated mainly from the patient’s blood, due to which the risk of immune rejection in the host body is almost eliminated [169]. Fibrinogen enzymatically polymerizes in the presence of thrombin to form fibrin hydrogels. This hydrogel has poor mechanical characteristics but promotes the differentiation of MSCs into chondrocytes and cartilage formation [197,198,242].

Plasma fibronectin is an evolutionarily conserved glycoprotein that is directly involved in cell interactions and engages in processes of cell adhesion, proliferation, motility, differentiation, opsonization, and apoptosis [204,210,243,244,245,246]. The structure of plasma fibronectin is a dimeric glycoprotein consisting of two subunits with a total mass of 500 kDa with variable molecular conformations and chemical binding variants [247,248,249]. Another function of plasma fibronectin is the participation of this adhesive high-molecular-weight glycoprotein in the processes of repair and healing of skin lesions [250], periodontal elements [251], bone [252], heart valves [253], the cornea [254], the mucosa of the tongue [255], and peripheral nerve trunks [256].

The possibility of plasma fibronectin usage as a growth factor is very perspective. Volume hydrogels based on plasma fibronectin with controlled stiffness and degradation ability are being developed. Such hydrogels include a full-length biopolymer, which provides the most physiological variant of the solid-phase presentation of growth factors. It was demonstrated in vitro and in vivo the effect of incorporating vascular endothelial growth factor (VEGF) and bone morphogenetic protein 2 (BMP2) into these hydrogels to enhance angiogenesis and bone regeneration, respectively [247].

In another recent study, a photocross-linkable plasma fibronectin conjugate was incorporated into tridimensional hyaluronic acid hydrogel networks to enhance endothelial cell adhesion and angiogenesis [202]. In addition, the authors proved that the combination of plasma fibronectin with hyaluronic acid in the hydrogel gives the latter the ability to retain water while increasing its mechanical properties since the plasma fibronectin molecule adds rigidity to the polymer network due to its significant molecular weight (~220 kDa) [204].

Hydrogels based on natural biopolymers, with all their advantages, also have drawbacks, including low mechanical properties and difficulty in modifying the chemical properties. Therefore, along with natural biopolymers, biopolymers of a synthetic origin are used in biofabrication due to the possibility of controlled adjustment of their mechanical properties and biochemical characteristics, as well as the ability to form compositions with other biomaterials, including biological ones, which can give certain advantages in tissue engineering of HAC [200,225,257].

To date, the most active in the biofabrication of HAC, including TDB, are hydrogels based on such synthetic biopolymers as polyethylene glycol and polyvinyl alcohol, and some others [257].

Polyethylene glycol (PEG) is a linear carbon polymer, based on the ethylene oxide oligomer, which can be photochemically cross-linked into a stable hydrogel. PEG is a well-studied and widely used synthetic biopolymer for hydrogels due to its properties such as nontoxicity, biodegradability, nonimmunogenicity, and good solubility in organic solvents and water. Among the disadvantages of this biopolymer is the lack of qualities such as bioadhesion. PEG-based hydrogels are applied in various studies in the field of tissue engineering of HAC due to their ability to maintain cell viability and stimulate the synthesis of cartilage ECM. In addition, PEG-based hydrogels can withstand both compressive and tensile loading well and can easily be modified with different functional groups such as carboxyl, thiol, diacylate, or acrylate. Thus, the biochemical and biophysical properties of PEG hydrogels can be specifically modified to produce HAC biosimilars with given characteristics [257,258,259].

Polyvinyl alcohol (PVA) is another representative of biopolymers widely used for biofabrication, including TDB of HAC biosimilars. PVA is a non-toxic and nonimmunogenic water-soluble biodegradable synthetic polymer. A large number of side hydroxyl groups not only makes PVA hydrophilic but also allows the biopolymer to form a semi-crystalline structure due to intramolecular hydrogen bonding [257,260]. PVA transforms into a hydrogel by chemical or physical cross-linking. Chemical cross-linking is conducted using functional cross-linking agents such as glutaric aldehyde. The physical cross-linking performance is with heating, mechanical action, or ionizing radiation [260,261,262,263]. The widespread and active use of PVA hydrogels in the regenerative medicine of HAC is due to their ability to exhibit similar mechanical and structural properties to natural cartilage ECM [260,264]. Difficulty of integration with the surrounding native cartilage tissue is the disadvantage of PVA [265,266], which may be eliminated by combining PVA with polylactic-co-glycolic acid (PLGA) [267].

Polylactic-co-glycolic acid (PLGA) is a copolymer of lactic and glycolic acids, is biodegradable and biocompatible, and has good water solubility [268,269]. Independent gelation of this biopolymer is difficult, so it is often combined with PVA and PEG. In regenerative medicine, PLGA scaffolds are also used in combination with platelet-rich autoplasma (PRP), which contains and releases growth factors to regulate the tissue healing process [270]. PRP also provides effective delivery of MSCs to PLGA scaffolds with subsequent cell proliferation and differentiation. A similar combination of PLGA and PRP finds application in bone and cartilage tissue engineering [271].

Polylactide-poly-lactic acid (PLA) and its copolymers belong to the group of hydrophobic aliphatic polyethers with lactic acid as the monomer. PLA hydrogels are applied in biofabrication of supporting tissues due to their adequate mechanical properties, biodegradability, biocompatibility, thermoplasticity, and shape memory effect [272,273].

Polycaprolactone (PCL) is a synthetic polyester produced by ε-caprolactone ring opening polymerization. PCL is applied in tissue engineering because of its low melting point, adequate biodegradability, biocompatibility, mechanical strength, and relatively low cost. This biopolymer is engaged in scaffold formation, including combinations with polyurethane (PU). Biofabrication of biosimilars of bone, muscle, cartilage, skin, and cardiovascular tissue actively involves PCL and PU [274,275].

Poly(ethylene oxide)-b-poly(propylene oxide)-b-poly(ethylene oxide) or pluronic, is a heat-sensitive synthetic biopolymer capable of gelation. Pluronic in combination with some other biopolymers is capable of imparting unique and valuable qualities to materials for biofabrication, including TDB. For example, a hydrogel based on chitosan and pluronic showed more effective chondrocyte proliferation and increased expression of ECM compared to an alginate hydrogel [276].

Hydrogels created as materials for TDB can contain biopolymers of a biological or synthetic origin, as well as their various combinations, depending on the necessity of achieving required qualitative characteristics. Complex hydrogels may have the following combinations of biopolymers in composition: gelatine/PLGA [277]; PCL-alginate [278]; polyurethane/hyaluronic acid [279]; alginate with the natural biopolymers of gelatine, agarose, collagen, hyaluronic acid, etc.; and alginate with the synthetic biopolymers of polyethylene glycol, pluronic acid, etc. [168].

Hydrogels with simulated properties. Based on the understanding that during the development of hydrogels for TDB, there are a huge number of factors to be considered (selection of the biopolymer, solvent, additional hydrogel components, and consideration of internal and external conditions of gel formation), the use of automated systems for modelling and predicting future characteristics of hydrogels becomes optimal. Thus, authors have developed a computational model of the hydrogel polymer network, which provides a fast and accessible strategy for predicting the properties of the biopolymer network (polymer content, monomer composition, the polymer chain radius, the cross-link density, and the cell size) and, ultimately, developing on its basis hydrogel systems with desired properties for potential therapeutic applications. The high efficiency and validity of the developed computational model for a wide range of alginate polymers and hydrogels based on them have been confirmed [280].

Hydrogels developed for cartilage and bone tissue repair may also include such components as living cells, the mineral–organic composition of cartilage tissue, proteoglycans (protein–polysaccharide molecules) [281], noncollagen proteins, glycoproteins (fibronectin, ankyrin CII, tenascin, and cartilage oligomeric protein), metabolites, etc. [166]. Additionally, to control the viscosity of hydrogels, various gelatinizing agents can be added, such as combined biopolymer carboxymethylcelluloseem [161,179].

##### Bioinks

Another basic material for biofabrication with TDB is bioinks—cytocompatible hydrogels based on synthetic and natural biopolymers, which include cell culture—preferentially in the form of spheroids and auxiliary components—drugs and signalling molecules (hormones, cytokines, growth factors, and neurotransmitters). In this case, the hydrogel plays the role of an ECM analogue, which creates the necessary conditions for the growth, differentiation, and proliferation of cells depending on the purpose (area of application) of the bioinks [170,219,282].

In terms of cell culture, as one of the components of bioinks, it is optimal to use chondrocytes, mature cells that have preserved the ability to divide and are the basis of cartilage tissue, for biofabrication of HAC. Chondrocytes can be isolated from the donor sections of the patient’s own HAC and then cultured to obtain the required number of cells sufficient to produce bioinks. Studies have shown the successful use of chondrocytes in tissue engineering, including as part of bioinks for cartilage’s analogues’ TDB [178].

Unfortunately, applying chondrocytes has some disadvantages, such as traumatization of the donor site and the high cost of cultivation. In the first case, despite the intake of chondrocytes from the lateral, load-free surfaces of the HAC, the resulting artificial defects can spread to the compression, that is, the surface of the joint that experiences pressure and motion, aggravating the course of the disease or injury and increasing the defect zones. In the second case, the high cost of bioinks’ preparation, due to the procedure of cell culturing, turns cartilage TDB into an economically inexpedient treatment option. Therefore, researchers have proposed alternative allogeneic and autologous cell cultures for bioinks’ production and following usage in the HAC’s biosimilar TDB. These are progenitor or induced pluripotent stem cells, bone marrow-derived MSCs, adipose tissue, the synovial sheath, chondroblasts, and fibroblasts [181,183,206,207,213,222,283]. MSCs and progenitor cells as well as chondroblasts can be differentiated along the chondrogenic lineage by applying growth factors. On the other hand, there are difficulties related to the directed differentiation of the mentioned cells and, as a consequence, the tendency to differentiate hypertrophic [161,185]. So, the search for the optimal cells’ sources in the aspect of bioinks’ production as a material for HAC analogues’ TDB continues.

Applying multiple cell types or cartilage-differentiated cells at various stages of differentiation as a part of bioinks could be a serious potential in the biofabrication and TDB of HAC. So-called co-culturing demonstrates the potential to enhance the chondrogenic properties of the finished tissue-engineered construct. In particular, a similar effect was noted when using MSCs and chondrocytes [231], as well as chondroblasts and fibroblasts with chondrocytes [173].

Stem cells, especially MSCs, are promising because of the potential in the chondrogenesis and cartilage regeneration [200]. MSCs can be isolated from various sources, such as the bone marrow, adipose tissue, synovial sheath, periosteum, skeletal muscle, skin, amniotic fluid, and umbilical cord blood [284]. The most accessible and technologically simple method is MSC obtainment from adipose tissue. The MSCs from this source have shown good clinical prospects for HAC restoration [252]. MSCs have a high regenerative potential and are capable of differentiating into several age-related cell cultures, as well as influencing differentiation through the release of growth factors and cytokines with the impact of the latter on the nearby tissues. In addition to cellular regulators of HAC formation, a wide range of biopolymers and hydrogel matrices based on them that promote chondrogenesis have been proposed and applied. Such matrices mimic the extracellular environment of chondrocytes, which promotes chondrogenic differentiation of chondroblasts and MSCs [200,283]. Chitosan, collagen, alginates, and gelatine are most commonly used as biopolymers for creating such matrix bases [193,283,285]. Gelatine is often chemically modified, for example, with gelatine-methacryloyl groups. The resulting modified gelatine may be combined with chondroitin sulfate or hyaluronic acid. Such combinations are created due to the existing disadvantages of gelatine. Thus, in a recent study, bioenzymatic non-sulfated chondroitin was tested for the first time in a synthesized chemically cross-linked matrix based on gelatine, proving its effectiveness as a potential basis for use in the field of cartilage regeneration, and the hydrogels developed on the basis of this combination confirmed the ability to maintain the viability of MSCs [200].

Growth and cell proliferation factors can also act as auxiliary components of bioinks. Growth factors are the protein molecules that regulate cell division and proliferation (they are often polypeptides that stimulate or inhibit the proliferation of different cell types) [286]. The growth factors that are the first to trigger the cascade of bone and cartilage regeneration processes include platelet-derived factors (PDGF), transforming growth factor beta (TGF-β) from immunocompetent cells, and fibroblast growth factor (FGF). In particular, autogenous PRP-gel with growth factors is actively used in biofabrication to improve the healing of supporting and connective tissues [286,287].

In another study, the authors developed a new heparinized synthetic carrier based on photo cross-linked methacrylate glycol–chitosan and conjugated heparin. Such a carrier was applied to form a stable high-molecular-weight complex to enhance the bioactivity of bone growth factors [196]. The heparin usage makes it possible to create biomaterials with controlled release of heparin-binding proteins, such as TGF-β1 and morphogenetic protein. On the other hand, heparin also influences cell migration, proliferation, cartilage differentiation, and synthesis of the HAC-specific ECM, as well as suppressing the immune response [288].

In a big prospective study, bone morphogenetic protein-4 (BMP-4) was used as a component of hydrogel with high potential to stimulate chondrogenesis, accelerate the repair of bone–cartilage defects, and preserve the cartilage structure after regeneration [178]. Additionally, another study developed innovative hydrogel replicates in a growth factor-enriched microenvironment to improve regeneration, which included bone morphogenetic protein-2 (BMP-2) [289]. BMPs act on cell membrane receptors and regulate the growth, differentiation, and apoptosis of various cell types including osteoblasts, chondroblasts, nerve cells, and epithelial cells [178]. The synergic effect of bone morphogenetic protein with ECM on osteogenic progenitor cells is an important osteoinductivity factor, promoting the differentiation of the osteoblast–osteocyte complex [289,290,291].

Hyaluronic acid, which can cross-link with other polymers and trap drugs and growth factors to achieve a controlled release, is used to stimulate cellular activity as well as the microenvironment of ECM, such as adhesion and proliferation [35]. In a conducted study, the authors created hybrid products by coaxial printing with bioinks produced using hyaluronic acid and a thermoplastic polylactic acid polymer [32]. Applying such types of hybrid products provides an optimal condition for chondrocyte growth as well as maintaining the mechanical properties necessary to withstand the stresses acting in vivo [32].

##### Quality Assessment Methods for Hydrogels, Bioinks, and Bioprinted Products

In the process of obtaining and developing hydrogels and bioinks and creating volumetric constructs in the TDB process, it is necessary to analyse their composition, physical and chemical properties, and the nature of the interaction between biopolymers, and assess the state and stages of development of cell components, as well as the subsequent study of stability and bioresorption of bioprinted products (bioimplants) after implantation, including their impact on the morphology and biochemistry of damaged surrounding tissue. For these purposes, a set of various physical and chemical research methods are used to assess the structure and component composition of hydrogels and bioinks, among which there is infrared spectroscopy [292,293,294], Raman spectroscopy [295], the circular dichroism method [296], the fluorescence method [297], etc. In addition, electron microscopy is used to analyse the molecular organization of networks of biopolymers that form a gel [180,296,298].

Infrared spectroscopy with the Fourier transform (FTIR) is a technique that can provide important information about the self-assembly process leading to the formation of supramolecular hydrogels [296,299]. By analysing the wave number or intensity shifts, the involvement of specific bonds in the construction of the hydrogel network can be assessed [296]. Comparison of the infrared spectrum in the hydrogel state with the spectrums of solutions or solid states of gel-forming agents can provide information on the interactions governing the formation of the hydrogel network. Various types of non-covalent interactions are involved in the formation of the supramolecular hydrogel network: hydrogen bonds and van der Waals forces [296]. In many cases, gel-formers have functional groups in their structure with characteristic vibrational bands (carboxyl, hydroxyl, amino, and amide groups), which are sensitive to the formation of weak physical interactions [296].

Published results show the importance of using the FTIR method in the analysis, study of physical and chemical properties, and standardization of hydrogels [300]. Thus, the possibility of assessing the presence of albumin in hydrogel matrices using FTIR has been illustrated. Additionally, FTIR spectroscopy has been used in the study of hydrogel swelling under in situ conditions, and the FTIR images obtained in this work help characterize molecular processes in chemically sensitive materials [299]. 

FTIR was also used to evaluate bioimplants made from a new type of non-functionalized soft calcium alginate hydrogel in a model of spinal cord hemisection in rats. Using FTIR, the authors evaluated the stability of the implants and their effects on the morphology and biochemistry of the damaged tissue 1 and 6 months after injury [293]. In the current work on the study of mucin glycoprotein and the synthesis of methacryloylmucin hydrogels, the chemical modification of mucin and the characteristics of the resulting product were evaluated by infrared spectroscopy [294].

Combined dispersion spectroscopy or Raman spectroscopy is used to study the interactions within the mesh of biopolymers, to study the type of cross-linking and its density, to analyse conformational changes of biopolymer mesh under various influences in the hydrogel, and to later monitor in cell cultures, which allows studying the vibrational energy of molecules [296,301]. This method of investigation makes it possible to evaluate the structural organization or the characteristics of the diffusion properties of dissolved substances in hydrogels. Among the various Raman methods, surface-enhanced combined dispersion is characterized by the highest sensitivity [302].

Circular dichroism spectroscopy (CDS) is a valuable technique for studying the structure and dynamics of peptides, proteins, and nucleic acids, including those incorporated into hydrogels. CDS characterizes molecular conformational changes in real time. The method allows for high accuracy, sensitivity, and a good signal-to-noise ratio. Although CDS does not provide a detailed structure at the atomic level, it allows for evaluating the behaviour of biopolymer molecules and their interactions in hydrogels. Researchers use CDS to observe molecular phenomena, namely how macromolecules unfold/rearrange and how their overall self-assembly/disassembly occurs [296].

Confocal microscopy (CM) is a method of optical microscopy with high contrast through the use of an aperture that cuts off the flow of background scattered light [208,303,304]. The CM method is actively used to evaluate hydrogels and membranes obtained on its basis [305]. The peculiarities of the method are some advantages: with a CM, it is possible to study tissues at the cellular level in the state of physiological activity and to evaluate the results in four directions—height, width, depth, and time, and the use of a special attachment makes it possible to study at different temperatures [208,306].

Fluorescence spectroscopy (FS) can be used to evaluate the characterization of hydrogel systems containing fluorophore particles capable of absorbing energy, which allows excitation from the ground electronic state to one of the vibrational states in the excited electronic state. The method is used to monitor gelation properties [296]. An interesting example of the FS use in conjunction with CM is presented in a study of the mycotoxin citrinin in the condensed phase and hydrogel films of agarose and alginate. A fluorescence spectrum and attenuation analysis were used to recognize the presence of citrinin in hydrogel films [303]. In a promising study, the authors provide evidence for the ability of hydrogels to function effectively as drug carrier systems in vitro using various cancer cell cultures for 7 days, and the study itself shows that relatively simple spectroscopic measurements of FS contribute to a fundamental structural and chemical understanding of the properties of protein hydrogels [307].

### 3.2. Bioprinting Stage

#### Bioprinters and the Bioprinting Process

The main stage of creating scaffold-based and scaffold-free tissue-engineered constructs by the TDB method is implemented in 3D bioprinters—complex technical devices controlled by special software [155]. The main units of such devices are represented by a frame or case, cartridges with biomaterials for printing, a drive capable of moving in three planes, and a platform on which the forming of the construct is performed [95,96]. Modern 3D bioprinters are presented as technically simpler devices that have a limited set of components and options and are designed to work with one or two types of biomaterials, and complex multi-component units that can print using several variants of biomaterials—from metals and solid organic compounds such as hydroxyapatite to soft hydrogels and bioinks. In the latter case, the bioprinter implements several 3D printing technologies—sintering, electrospinning, extrusion, etc., and many modes of operation—thermal, baric, and others. Such a complex device can be equipped with several cartridges (from two to six) for simultaneous or coordinated alternate printing with different biomaterials (Figure 4). Additional options for the 3D bioprinter can be:Sterile printing chamber with the option of a bioreactor for functionalization of the printed construct in the post-bioprinting stage;Laser sensor system for automatic calibration of printing elements;Interchangeable platforms (or worktables) and substrates for bioprinting;Camera for photo and video recording of the manufacturing process and its control.

Other equipment can be added to the above. Currently, there are scientific and technical centres and companies that manufacture 3D bioprinters for scientific research and practical use, including commercial implementation [155,308,309].

The main TDB techniques, depending on the design of the 3D bioprinter, include the microextrusion, inkjet, and laser [95,219,309,310]. Microextrusion TDB uses the principle of automated mechanical extrusion (squeezing) of material from a cartridge, realized through one of two mechanisms: semi-solid extrusion and fusion modelling. In the first one, pressurized air or a piston (rotating screw) mechanism squeezes a continuous stream of semi-solid printing materials (such as hydrogels and bioinks) through the printhead nozzle, which is applied to the substrate layer-by-layer and fabricates a 3D object. In the second process, a thermoplastic biopolymer filament is fed into the nozzle of the printhead, where it is heated until it melts and then applied to the substrate and in layers to form a volume construct (Figure 5). The flexibility of the microextrusion process and the availability of materials make it the most commonly used TDB method [161,311,312].

The 3D bioprinters used in this TDB method have a temperature-controlled material handling and dispensing system and a heated/cooled substrate on which the construct is created. Both the cartridges and the substrate can move in all three Cartesian coordinate axes under the control of a positioning system. Some microextrusion bioprinters use multiple printhead cartridges (up to six in modern devices), which can be equipped with a heating and cooling system to convert biopolymers and some hydrogels and bioinks into a semi-solid state suitable for extrusion. This allows simultaneous and coordinated handling of several materials at the same time. In addition, an ultraviolet (UV) light source for photocross-linking biocompatible oligomers and a dispenser for spraying solutions of chemical cross-linking agents may be installed in the printing chamber [155,312].

Microextrusion TDB technology finds its application for the fabrication of structures, including hybrid cell–tissue constructs with a high level of stratohistological differentiation, such as HAC biosimilars with a subchondral bone layer, bone tissue with bone–tendon–muscle blocks [313,314,315].

Inkjet (injection) TDB is one of the most common and technically simple options for creating bioprinted constructs. Initially, inkjet technology was widely engaged in two-dimensional printing using conventional chemical inks. Later, it was modified by giving the ability to the cartridge of being moved in three dimensions, replacing the ink inside with biomaterial, and using a special substrate instead of paper (Figure 6). Currently, 3D bioprinters for inkjet TDB while working with biomaterials can provide high accuracy, speed, and resolution of printing. In addition, this technology is characterized by a minimal percentage of cell damage. Inkjet TDB finds its application in regenerative medicine for cartilage and skin injuries [316,317,318].

In addition to continuous inkjet printing, 3D inkjet bioprinters use piezoelectric or thermal force development systems to form individual droplets of material in liquid form and then outputs them to a special substrate on which the volume construct is formed layer-by-layer. Both approaches allow for consistent flow or drop-by-drop (“drop by demand” technique). In the piezoelectric drop formation activation method, the electrical signal from the piezoelectric sensor is converted into a mechanical force and lets the material be ejected from the printhead in individual droplets. The thermal activation method results in local heating of the material, converting it to a vapour state that creates enough pressure to release the liquid by droplets [319]. All these approaches have advantages and disadvantages, so the choice of a particular technology will be determined by the characteristics of the biomaterial and the tasks at hand. For example, local pressure can damage the MSCs as they leave the cartridge. Therefore, optimization of the value of the mechanical force will ensure maximum preservation of the cell’s component [320].

One of the limitations of inkjet TDB is that the printing materials (hydrogels and bioinks) must be in liquid form. Accordingly, the fabrication of a complex volumetric structure with shape preservation during and after the printing process becomes quite problematic. To offset this limiting factor, advanced technological approaches can be used, namely hybrid printing (use of temporary or sacrificial elements, reinforcing layers, and lattice structures made of bioresorbable material created by the extrusion TDB method). The disadvantages of the inkjet TDB method include the inability to use hydrogels with high viscosity, frequent sticking, and deposition of cells in the cartridge, and clogging of the nozzle holes of the print head [195,321].

The process of microextrusion or inkjet TDB has a certain difficulty in ensuring optimal conditions to preserve the viability of the cell spheroids introduced into the hydrogel matrix. Two technological approaches have been developed to solve this problem. The first involves the use of a needle-shaped coaxial nozzle on the printhead. The material flows in the inner and outer channels of the nozzle and converges at the end. The flow rate in each channel can be independently controlled to give the designs optimal characteristics [322,323,324]. Coaxial TDB allows the creation of complex tissue constructs by controlled deposition of biomaterials. Using the coaxial TDB, volumetric constructs with osteoblasts and endothelial cells were created by depositing angiogenic and osteogenic bioinks from the central and coaxial nozzles, respectively [324,325]. The essence of the second approach is the use of a special confluent nozzle with which hydrogels, bioinks, and cytocompatible biopolymers can be printed synchronously with the formation of a combined print thread in the confluent area of the nozzle, which seriously simplifies the printing process, but carries the risk of cell damage [326].

Originally developed for 3D printing with metals, direct laser-induced transfer technology was later successfully applied to biological materials (hydrogels and cell spheroids with a culture medium, and single proteins) [327]. A laser 3D bioprinter based on DLP technology consists of a pulsed laser beam source, a moving ribbon of printing materials, and a receiving substrate located opposite the ribbon. The ribbon is double-layered, and the first or top layer has a liquid base, while the second layer contains a hydrogel with cell spheroids. When the first layer is exposed to laser light, the liquid evaporates instantly, and the resulting vapour presses the second layer. As a result of this pressure, a portion (drop) of hydrogel with cells is released from the second layer at a given location onto the receiving substrate. The receiving substrate contains biopolymers or a cell culture medium that promotes cell adhesion [328,329] (Figure 7). Laser TDB is currently used to create multilayer tissue–cell constructs of the skin with a given cell density and is characterized by high precision. Among the disadvantages of the technology are the high cost, low printing speed, and damage (death) of 5 to 30% of cells in the process of the laser-induced transfer [328,329,330,331].

The TDB process may also require the solidification of printed materials (biopolymers and hydrogels) applied to the substrate and then layer upon layer. This process is known as cross-linking. Various chemical and physical factors are used to force the fixation of the structure and shape of the tissue-engineered construct. Chemical ones include the addition of cross-linking regulators (calcium ions in the form of calcium chloride are sprayed onto alginate hydrogels) or mixing hydrogels and bioinks with various additives (fibrinogen and thrombin). Physical ones include ultraviolet (UV), changes in pH, pressure, temperature, and oxygen concentration. Depending on the stage at which TDB is affected by these factors, the following types of cross-linking are distinguished [112,311,332]:Pre-cross-link—processing the printed biomaterial in the cartridge;Post-cross-link—processing the printed biomaterial after it leaves the cartridge;In situ cross-linking is the processing of printed biomaterial in the nozzle area of the printhead.

The limitations of existing TDB methods and the shortcomings of the printing materials affect the quality and properties of the final tissue-engineered construct. Changing and improving its properties, as well as making the manufacturing process more controllable, are possible by using hybrid TDB technologies. Such an approach implies the engaging of different biomaterials and TDB technics in a single technological process to overcome the traditional limitations of individual technologies. At the outset, we obtain the most precise morphological reproduction of the natural biological qualities of tissues and organs. To date, the following options for the implementation of hybrid TDB are distinguished [333,334,335]:The usage of temporary (sacrificial) structures;Manufacturing of structures with internal reinforcement;Application of coordinated extrusion bioprinting;The usage of a modular assembly.

In the first case, the bioinks act as the core constructive material for the fabrication of a complex-shaped structure. Parallel biopolymers and some hydrogels based on them form temporary (sacrificial) supports of the structure at the fabrication stage and are subsequently removed. The sacrificial approach is typically realized through microextrusion TDB. Using it, precision TDB of complex anatomical shapes, such as a femoral head, an ear, and a blood vessel, has been performed. In doing so, biopolymers based on polycaprolactone, or sodium alginate, were used to form sacrificial structures and temporarily support the main structure from a hydrogel based on gellan gum and gelatine methacrylate [334].

Another variant of hybrid TDB using temporary materials is the so-called immersion printing method using special hydrogel baths [336,337]. The method is based on layer-by-layer printing of constructs by individual spheres (drops) of bioinks inside a hydrogel support base in both vertical and horizontal directions, which is especially important when creating structures with a branching structure (blood vessels) [338]. The immersion technology is devoted to the drawback in the form of difficulties with regulating the physical properties of the supporting temporary structures, which take place in the hybrid sacrificial TDB method described above. Moreover, the ideal hydrogel for creating an immersion bath should have such a shear flow limit to promote active cell proliferation in spheroids with the manufacturing of tissue constructs [339].

The prospects of using an immersed TDB in dense media have been demonstrated. In particular, authors propose a technology realized by immersing the printhead nozzle in high-density perfluorocarbons, which are chemically inert and have an excellent ability to transport oxygen and carbon dioxide, providing an ideal environment for immersed cell spheroids [340,341]. Other biomaterials are also used to form the immersion medium: perfluorotributylamine (a high-density hydrophobic fluid) [336] and gellan gum, in which a fully vascularized adipose tissue biosimilar has been printed [337].

Hybrid TDB with internal reinforcement of the structures is carried out by applying bioinks, as the main constructive material, to a prefabricated mesh frame made of biopolymers and/or hydrogels. Such a framework acts as a reinforcing mesh and significantly strengthens the hybrid structure. The TDB microextrusion method is used to implement this approach. In some studies, volume constructs printed with MSCs and hydrogels based on gelatine methacrylate or polyethylene glycol–diacrylate were reinforced with polycaprolactone biopolymer fibres [342,343]. The resulting reinforced structures had a higher mechanical strength (60–175 kPa), compared to those made only from bioinks (45 kPa). The increase in strength up to a certain limit was proportional to the content of the reinforcing fibre. It was also observed that cell spheroids interacted with reinforcing fibres by generating separate binding sites by the latter and causing focal adhesion [344,345].

The coordinated hybrid TDB involves, contrary to the previous technique, the sequential (layer-by-layer) extrusion of bioinks and biopolymers that reinforce the structure being created. This is achieved through the alternating work of the various print heads of the 3D bioprinter, whose cartridges contain different biomaterials for printing. The coordinated process is controlled by a computer program. The studies described coordinated TDB printing with polycaprolactone-based hydrogel and alginate-based bioinks containing MSCs. As a result, the authors obtained a cell-loaded structure with mechanical strength comparable to that of cartilage tissue (6 MPa). However, the cell survival rate using this approach was 60–80% [346].

Modular TDB implies the production of tissue micromodules with their subsequent assembly according to the constructor principle with the formation of larger and more complex functional units. Thus, tissue micromodules are primary elements—analogues of the building blocks of the future tissue-engineered structure. The modular approach is characterized by the high survivability of cells and allows the assembly of complex constructs, consisting of different types of cells, repeating the cytoarchitectonics of natural tissue [69]. A classic example of tissue micromodules is cell spheroids, placed inside supporting frameworks based on biopolymers of different shapes, most often hexagonal. The latter are then combined into structures of any complexity, and the cell spheroids contained within subsequently fuse and begin to produce the ECM, forming a new tissue [309,347].

### 3.3. Post-Bioprinting Stage

#### Bioreactors In Vitro and In Vivo

The post-bioprinting stage is implemented using special high-tech equipment called bioreactors. Bioreactors can be defined as devices in which the properties of a printed tissue-engineered construct change under the influence of environmental factors of mechanical, physical, chemical, and biological nature over a certain time until it can be used for surgery [60,68,348,349]. Thus, conditions are created for the development and maturation of tissue-engineered constructs in vitro, acquisition of functional characteristics through the transmission of regulatory signals of different natures to the cells, and stimulation of them to differentiation and ECM production before implantation in vivo. That is, a dynamic microenvironment is formed, which is necessary for the cells at different stages of maturation, with all biochemical or mechanical and physical processes developing and proceeding under strictly controlled and monitored conditions [350,351,352,353].

However, the best bioreactor for tissue-engineered constructs is a living organism. Any in vitro bioreactor can reproduce only a small fraction of the full functionality of an in vivo bioreactor [354]. In vitro-printed tissue-engineered constructs are characterized by a lack of a regenerative microenvironment, including a set of components of nervous, immune, and endocrine systems, as well as a lack of trophic and vascularization. As a result, the transplanted construct has a deficit of autonomous vascularization and innervation, which affects cell survival and tissue regeneration processes [355]. The principle of an in vivo bioreactor is based on the body’s capabilities to provide key components necessary for the construct integration and further regeneration, including stem cells and growth factors. For this purpose, a limited artificial space in the body tissues is formed where a scaffold with auxiliary components necessary for new tissue growth is placed. To saturate the scaffold and the space around it with cell culture (pluripotent or specific stem cells), to stimulate neohistogenesis, vascularization of the scaffold with a mobilized vascular loop is performed. Perfusion provides an opportunity for the recruitment of stem cells and obtaining nutrients from the host (patient) [356]. As a result, in the area where bioreactor in vivo was organised, new tissue formation occurs, which can be applied in regenerative medicine [357,358,359]. With regards to HAC injuries, this approach can be implemented at the subchondral bone level in the cases of deep and extensive cartilage defects.

## 4. Vectors for the Development of Tridimensional Bioprinting of Hyaline Articular Cartilage

Today, TDB is one of the promising methods of additive manufacturing and biofabrication of complex volume constructs with given rheology and increased structural, mechanical, and biological properties for organ and tissue regeneration [282]. One of the advantages of additive manufacturing technology is the ability to produce personalized implantable individual constructs considering the anatomy, pathology, and biomechanical properties of the patient’s tissues [282,360].

Biofabrication and TDB, including the creation of HAC biosimilars and articular cartilage defect replacements, have made an impressive step forward in their development over the last quarter century since their emergence. It appeals to all aspects of the TDB technology—equipment, appropriate software, biomaterials, and their compounding, as well as methods of printing and functionalization of bioprinted volume constructs. In experiments involving laboratory animals and, in some countries, in clinical trials, products of the required scale are already being used, capable of replacing damage to HAC and the underlying subchondral bone with subsequent resorption and replacement of fibrous and even full-fledged HAC within the process of reparative chondrosteogenesis [361]. However, these approaches are still far from being implemented in clinical orthopaedics.

Further development and improvement of TMB approaches of HAC biosimilars are associated with the usage of biomaterials capable, according to the results of 3D printing, of reproducing the structure of ECM and its composition (internal structure, pore size, stiffness, and protein composition, including morphogenetic proteins—BMP-2, BMP-3, BMP-4, BMP-6, BMP-7, and the growth factors TGF-β, PDGF, IGF I, IGF II, bFGF, and aFGF) with maximum precision. The optimal sources for the fabrication of such tissue-engineered constructs are biomaterials of an allogeneic origin. They are more compatible with the tissue environment than other biomaterials, have inductive and conductive activity, are not toxic, and meet the criteria of reparative regeneration in terms of resorption with substitution by the newly formed original tissue. Moreover, during the first stages of tissue-engineered construct existence, its matrix made of allogenic materials excellently plays a nutrient medium role for the cell spheroids contained therein, ensuring their proliferation and differentiation. Considering the influence of mechanical loading factors and oxygen concentration on the indicated processes in HAC, the stage of post-bioprinting with the tissue bioreactor’s usage gains particular interest to researchers.

A significant breakthrough in tissue biofabrication was the development of the technology of obtaining and creating decellularized ECM (dECM). However, a personalized approach to its use, as applied to the elimination of defects in the area of HAC and subchondral bone, turned out to be difficult. The way out was the transformation of dECM into the form of the hydrogel, using the latter to build volume constructs by the TDB method. However, until now, hydrogels made of dECM and 3D bioprinted constructs based on them cannot fully reproduce the complex structure and composition of native ECM of HAC, since the specific spatial position of each unique protein and even their composition are violated in the process of transformation of dECM into a hydrogel. Accordingly, improving the available physical, mechanical, and biochemical methods of creating hydrogels from allogenic biomaterials should also become a point of research efforts.

The hydrogels obtained from dECM are represented prevalently by collagen, the main protein of ECM of HAC, which, unfortunately, is characterized by low chondrogenic activity, weak mechanical strength, and significant shrinkage after TDB and implantation. To improve the mechanical properties and biological activity of the collagen-based volume matrix, its properties could be modified by mixing with other biopolymer molecules (e.g., chitosan or synthetic polymers). However, the possibilities of using additional materials are limited due to collagen easily denaturing under the influence of various factors. In this aspect, the concept of a hybrid TDB using temporary sacrificial structures, internal reinforcement, and modular assembly has good prospects. The configuration of hybrid scaffolds may be the optimal solution to improve the mechanical and structural components of 3D collagen-based dECM constructs without compromising their bioactivity and compatibility with cells.

The developmental trajectories outlined in this review will make it possible to achieve the required level of biomimicry and create anatomically and functionally relevant tissue-engineered constructs of HAC by TDB.

## Figures and Tables

**Figure 1 polymers-15-02695-f001:**
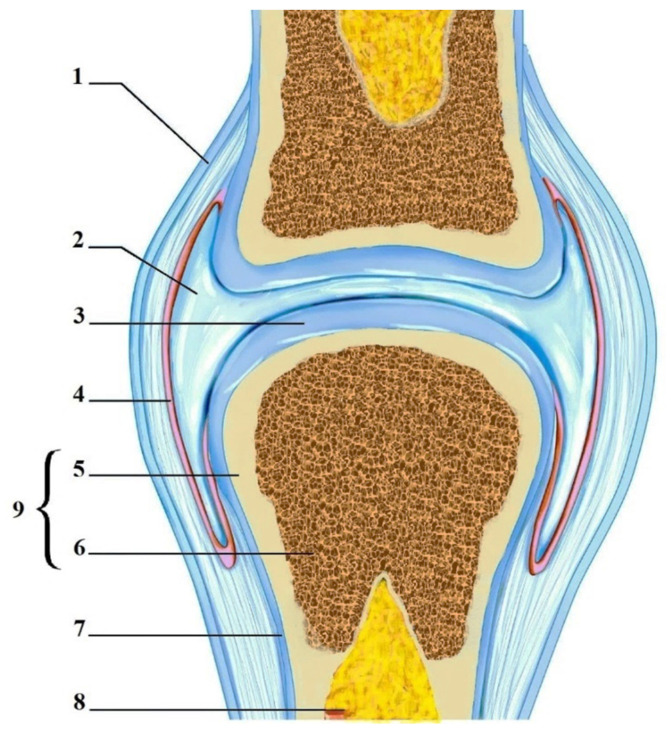
Articular cartilage. 1—Articular capsule; 2—Joint cavity with synovial fluid; 3—Hyaline articular cartilage; 4—Synovial membrane; 5—Compact bone; 6—Spongy bone; 7—Periosteum; 8—Bone marrow; and 9—Bone.

**Figure 2 polymers-15-02695-f002:**
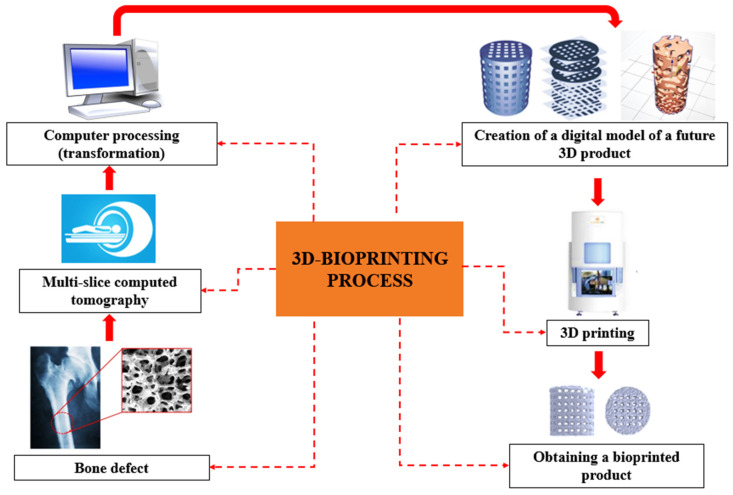
Visualization with modelling and 3D printing of a pre-product.

**Figure 3 polymers-15-02695-f003:**
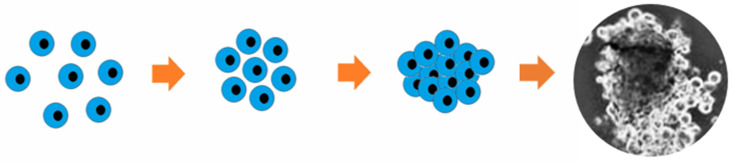
Stages of cell spheroid formation.

**Figure 4 polymers-15-02695-f004:**
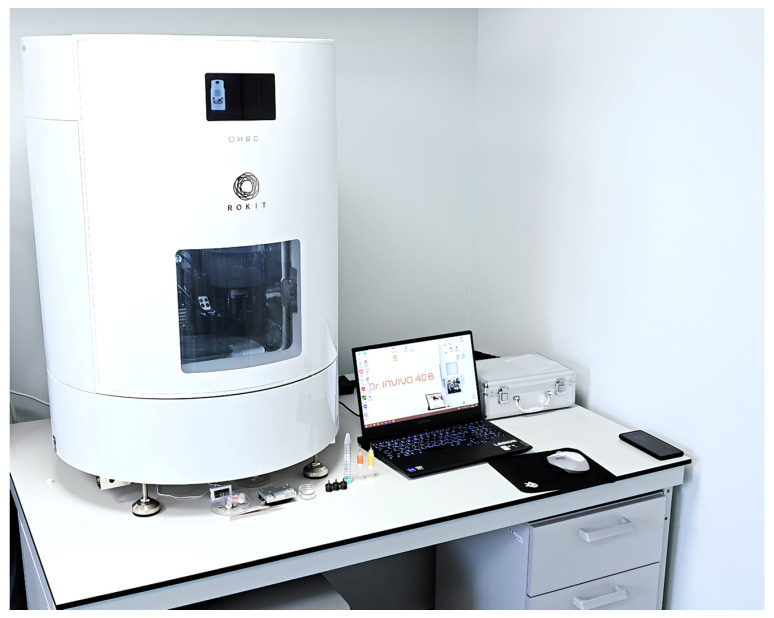
Photographic images of modern multifunctional 3D bioprinter.

**Figure 5 polymers-15-02695-f005:**
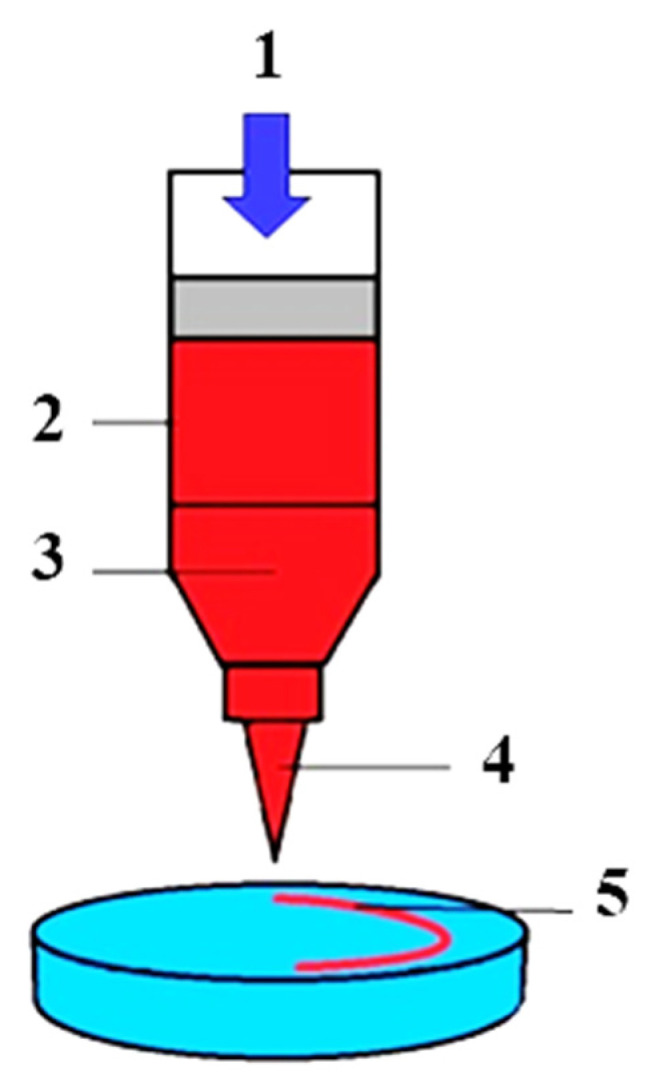
Schematic illustration of TDB technology based on microextrusion printing. Notation: 1—Piston; 2—Syringe; 3—Hydrogel; 4—Nozzle; and 5—Fabrication platform.

**Figure 6 polymers-15-02695-f006:**
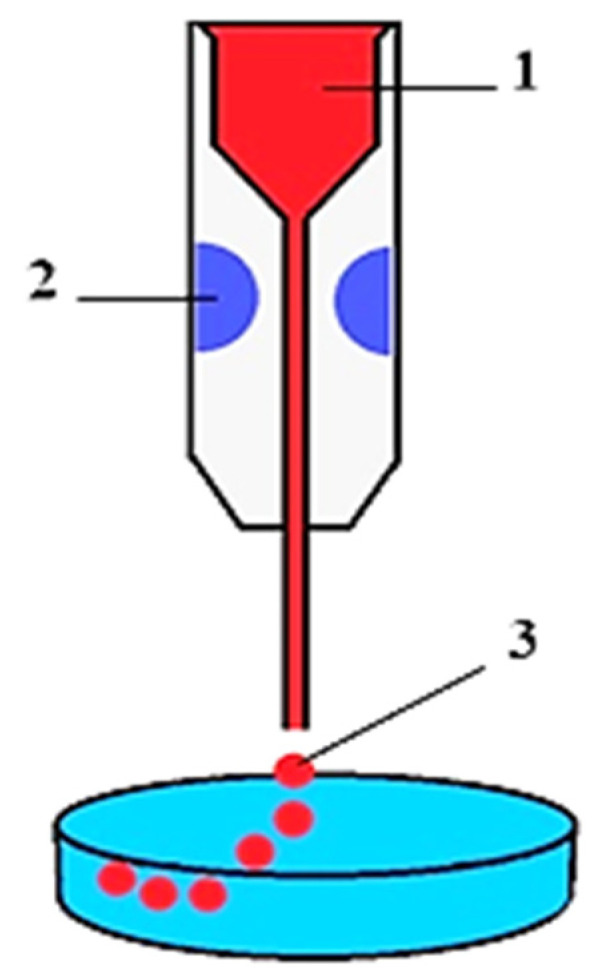
Schematic illustration of TDB technology based on inkjet printing. Notation: 1—Hydrogel; 2—Piezoelectric actuator; and 3—Droplet.

**Figure 7 polymers-15-02695-f007:**
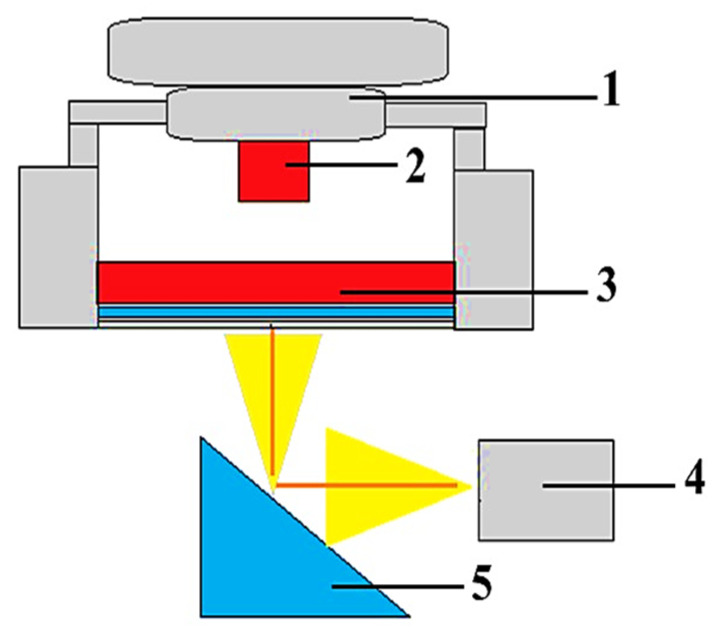
Schematic illustration of TDB technology based on laser printing. Notation: 1—Deposition platform; 2—Scaffold; 3—Hydrogel; 4—UV light; and 5—Digital micromirror device.

**Table 1 polymers-15-02695-t001:** Characteristics of the basic properties of hydrogels.

№	Properties of Hydrogels	Description of Properties	Sources Cited
1	Physical	Gelation temperature; gelation kinetics; rheology (flowability, viscosity, elasticity, and solidification/glazing ability); hydrophobicity/hydrophilicity; wettability; adhesion; polarization; and light transmission.	[151,154,163,165]
2	Chemical	pH; ion concentration; molecular weight of polymers; cross-linking ability; reactivity; and polarity.	[154,161,163]
3	Mechanical	Elasticity (ability to compress and swell); ultimate tensile strength; stress relaxation; and self-restoration and degradation.	[154,160,161,171]
4	Morphological	Porosity of structure and properties and size of pores.	[161]
5	Biological properties	Biocompatibility; cytocompatibility; biodegradability; inductivity; conductivity; and absence of irritant, toxic, inflammatory, and carcinogenic effects on surrounding tissues.	[154,160,161,170,171]

## Data Availability

Not applicable.

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
