# Peer review of "3D Bioprinting of Hyaline Articular Cartilage: Biopolymers, Hydrogels, and Bioinks"

_polymers, 2023, doi:10.3390/polym15122695_

Round 1
Reviewer 1 Report
In the review: “BIOPOLYMERS, HYDROGELS AND BIOINKS IN 3D-BIOPRINTING OF HYALINE ARTICULAR CARTILAGE”, the authors discussed about the importance of development of new treatment approaches for cartilage regeneration.
Overall, this manuscript results very interesting, the authors clearly explain the rational of the study and discussed the topic point by point.
However, we would like to invite the authors to clarify some minor points:
1. Please check the check punctuation and spaces;
2. Figure 1; Please try to modify the figure in order to see more details;
3.Maybe should be useful also describe the no-pharmacological approaches such as hyaluronic based viscosupplementation;
4. Within the description of scaffold and the use of mesenchymal cells also some in vitro study should be introduced. Please try to better describe the differences scaffolds available, based on natural and no natural polymers. In this respect the following reference should be useful: Vassallo V, Tsianaka A, Alessio N, Grübel J, Cammarota M, Tovar GEM, Southan A, Schiraldi C. Evaluation of novel biomaterials for cartilage regeneration based on gelatin methacryloyl interpenetrated with extractive chondroitin sulfate or unsulfated biotechnological chondroitin. J Biomed Mater Res A. 2022 Jun;110(6):1210-1223. doi: 10.1002/jbm.a.37364. Epub 2022 Jan 28. PMID: 35088923; PMCID: PMC9306773;
5. there are images of regenerated cartilage as examples?
56. In conclusion what we can say? What should be the best solution? There available data ?
minor revision of spelling is required
Author Response
Deeply esteemed Sir, we thank you for reviewing our manuscript and express our gratitude for your thorough analysis. Based on your objective comments and suggestions, we have corrected the material and supplemented it, which we hope has improved its content and informativeness.
Allow us to provide answers to your questions.
- We have reviewed and analyzed the article with a native English speaker, a specialist in biochemistry and microbiology, as well as by the built-in instruments of Microsoft Word. Excessive spaces, inappropriate punctuation (commas and semicolons) was removed. Thank you for this important note on the design.
- You are definitely right about the quality of the visualization of the former figure #1. We have replaced it with a larger and clearer figure with maximum detail.
- We apologize that we did not reflect in our review such a current approach as viscosupplementation in the treatment of joint diseases. We are very grateful for this reminder. We have added necessary information taken from a large profile review in the first section of the manuscript.
- We have reviewed the article you indicated and agree that we have unfairly omitted important information shown therein. We have included it in the list of references and, due to the contained information, we have better highlighted the information regarding the difference between natural and synthetic biopolymer-based available hydrogels and described the application of usage of scaffolds with MSCs (in vitro).
- Deeply respected reviewer, such images exist, but they are the authors' own and are an integral part of the results of preclinical and clinical trials of our colleagues. In this review, we did not use the author's images. Once we have the results of our own preclinical and clinical trials and have taken appropriate photos, we will be more than happy to present them in a future manuscript.
- We believe that the most promising approach is to use a hybrid 3D-bioprinting of hyaline articular cartilage biosimilar using dECM collagen as a matrix base enhanced with other biopolymers, as reported in the last paragraphs of the last section. At the same time, there are no data unambiguously proving the advantage of this or that type of bioprinting of articular cartilage analogues and profile studies in this direction are still actively pursued.
Let us once again express our gratitude to you. We will be grateful for future reviews!
Regards, authors’ team.

Reviewer 2 Report
In this review, authors discussed bioprinting of hyaline articular cartilage, from pre-printing to post-printing. Different materials and different types of bioprinter were mentioned. The idea to have such a review on this topic is good, which may contribute to the development of bioprinting. The summarize of different bioinks, cells and bioreactors would be a great guidance for readers in this field. This review is well organized with abundant analysis and citations. Please see some detailed comments below.
1. For the title, it seems hydrogel is included in biopolymer, and they are included in bioinks as well.
2. In abstract, since 3D was claimed for three-dimensional as abbreviations, it should be used after that.
3. Figure 2 is a little confusing, it’s difficult to tell the meaning directly from the figure itself.
4. For the classification of bioprinter, what is laser bioprinter? Is it DLP? For Figure 5, since these are different types of bioprinting, it’s better to explain and make captions one by one.
5. Please cite more latest papers in the manuscript.
English is good overall, only moderate editing is needed.
Author Response
Deeply esteemed Sir, we thank you for reviewing our manuscript and express our gratitude for your thorough analysis. Based on your objective comments and suggestions, we have corrected the material and supplemented it, which we hope has improved its content and informativeness.
Allow us to provide answers to your questions.
- Thank you for your absolutely correct comment regarding the wording of the title of the manuscript. We have transformed the title and it is now more logical, understandable and adapted to the content.
- We fully agree with this clarification. We have made corrections along the way, replacing three-dimensional with 3D.
- You are certainly right about the content and clarity of the former Figure #2. We have replaced it with a better and clearer illustration.
- You are absolutely correct about the technology behind laser bioprinting, which is DLP. We have added a reference to it in the corresponding paragraph of the manuscript.
- In this case, we also completely agree with you. Figure #5 has been divided by us into three separate illustrations.
- Deeply respected colleague! Thank you for your thorough analysis of the reference list. In the chronological structure of the list, 10% of the articles were published in the last 2 years, 20% in the last 5 years, and 50% in the last 10 years. We absolutely agree that the sources of information may be more "young". Let us correct this observation in one of our next manuscripts!
- Native English speaker, a biochemistry and microbiology specialist, made a review of the quality of the English language in the article and then did the necessary corrections.
Let us once again express our gratitude to you. We will be grateful for future reviews!
Regards, authors’ team.
